# Pharmacological Thromboprophylaxis in People with Hemophilia Experiencing Orthopedic Surgery: What Does the Literature Say in 2023?

**DOI:** 10.3390/jcm12175574

**Published:** 2023-08-26

**Authors:** Emerito Carlos Rodriguez-Merchan

**Affiliations:** Department of Orthopedic Surgery, La Paz University Hospital—Idipaz, 28046 Madrid, Spain; ecrmerchan@hotmail.com

**Keywords:** hemophilia, deep vein thrombosis, venous thromboembolism, pharmacological thromboprophylaxis

## Abstract

This narrative review of the literature, consisting of papers found in PubMed and The Cochrane Library published up to 31 July 2023, analyzed those that were deemed to be closely related to the title of this paper. It was encountered that the peril of deep vein thrombosis (DVT) in people with hemophilia (PWH) after orthopedic surgery is very small, such that pharmacological thromboprophylaxis is not necessary in most cases. The hemophilia literature states that the use of pharmacological thromboprophylaxis should only be performed in PWH undergoing major orthopedic surgery (total-knee arthroplasty, total-hip arthroplasty, ankle arthrodesis) who have additional venous thromboembolism (VTE) risk factors, such as old age, prior VTE, varicose veins, general anesthesia, cancer, factor V (Leiden) mutation, overweight, and treatment with the oral contraceptive pill (in females with von Willebrand’s illness). If we notice various risk factors for VTE in PWH who experience orthopedic surgery, theoretically, we should perform the identical type of pharmacological thromboprophylaxis advised for non-hemophilia patients: low-molecular weight heparins (LMWHs), such as enoxaparin (40 mg subcutaneous/24 h); or direct oral anticoagulants (DOACs), either thrombin inhibitors (dabigatran, 150 mg oral/12 h) or activated factor X (FXa) inhibitors (rivaroxaban, 20 mg oral/24 h; apixaban, 5 mg oral/24 h), or subcutaneous fondaparinux (2.5 mg/24 h subcutaneously). However, the review of the literature on hemophiliac patients has shown that only a few authors have used pharmacological prophylaxis with LMWH (subcutaneous enoxaparin) for a short period of time (10–14 days) in some patients who had risk factors for VTE. Only one group of authors used a low dose of DOAC in the dusk after the surgical procedure and the next day, specifically in individuals at elevated risk of VTE and elevated risk of bleeding after the surgical procedure.

## 1. Introduction

In 2023, Morfini et al. stated that, according to the Italian registry of surgery, hemophilia B individuals experience total joint arthroplasty less frequently than hemophilia A individuals. Additionally, they updated the hematological replacement therapy required for surgery utilizing continuous infusion versus bolus injection of standard and extended half-life Factor VIII (FVIII) concentrates in individuals with severe hemophilia A. They mentioned two important complications of surgery: venous thromboembolism (VTE) and the impact of inhibitors. Regarding primary hematological prophylaxis, for young hemophilia A patients, they recommended use three times or twice weekly, even with standard half-life (SHL) recombinant FVIII (rFVIII) concentrates. For patients with severe or moderate hemophilia B, they advised once weekly [1].

In the general population, VTE, i.e., deep vein thrombosis (DVT) and pulmonary embolism, is an uncommon adverse event after orthopedic surgery. The period of highest risk for VTE is the first 2 weeks after surgery [2]. The frequency of this complication in the general population has diminished considerably after the utilization of pharmacological thromboprophylaxis. However, there is concern about the increase in bleeding complications caused by pharmacological thromboprophylaxis when it is too aggressive [3]. Low-molecular weight heparins (LMWHs), such as enoxaparin (40 mg subcutaneous/24 h), have long been the pharmacological thromboprophylaxis commonly used in the general population. However, direct oral anticoagulants (DOACs), either thrombin inhibitors (dabigatran, 150 mg oral/12 h) or activated factor X (FXa) inhibitors (rivaroxaban, 20 mg oral/24 h; apixaban, 5 mg oral/24 h), or subcutaneous fondaparinux (2.5 mg/24 h subcutaneously) are increasingly being used as an alternative to LMWHs. In the general population experiencing orthopedic procedures, pharmacological thromboprophylaxis with DOACs has been found to reduce severe VTE and DVT significantly compared to thromboprophylaxis with LMWHs. However, the safety of both alternatives (DOACs and LMWHs) is similar [4]. Fondaparinux sodium (2.5 mg subcutaneous/24 h) has been reported to be more effective than LMWHs in hospitalized patients who have undergone major orthopedic surgery or trauma, especially in those who are elderly and have renal insufficiency or hypertension [5]. Rivaroxaban is also considered a good option to prevent DVT [6]. However, it is critical to remember that, despite pharmacological thromboprophylaxis, VTE can still occur [2].

## 2. Thromboprophylaxis in the General Population (Non-Hemophilia Patients)

### 2.1. Caprini Score

The peril of thrombosis can be calculated based on certain patient-specific factors and the surgical intervention utilizing the Caprini score, which is a validated model of VTE risk assessment [7] (Table 1).

For performing pharmacological thromboprophylaxis in the general population, Bartlett et al. (2020) have published a series of recommendations, shown in Table 2 [8].

For patient groups in which the Caprini method has not yet been proven (as is the case with orthopedic surgery), Bartlett et al. have recommended performing pharmacological thromboprophylaxis based on the risk factors of each individual and those specific to the surgical intervention they have undergone. The authors also recommended prolonging pharmacological thromboprophylaxis until the patient is able to walk or is discharged from the hospital; however, in some orthopedic surgery procedures, the duration of pharmacological thromboprophylaxis may be prolonged [8].

### 2.2. Major Orthopedic Hip and Knee Surgery

It is judged that 40–60% of the general population experiencing major orthopedic surgery of the hip and knee who do not receive pharmacologic thromboprophylaxis will develop DVT. To prevent DVT, rivaroxaban is a good option [6]. A study compared the efficacy of aspirin prophylaxis (PO 100 mg once daily) after knee surgery with the efficacy of LMWHs, vitamin K antagonists [oral warfarin, begun the dusk after the surgical procedure with the dose titrated to accomplish an international normalized ratio (INR) of 2–3, and factor Xa inhibitors (apixaban, rivaroxaban, fondaparinux)]. No substantial dissimilarities were found between aspirin, LMWHs, and warfarin in terms of their efficacy in preventing VTE. Factor Xa inhibitors were also encountered to be more efficacious than the other drugs in preventing VTE, although they also produced more bleeding complications [3].

### 2.3. Knee and Hip Arthroplasty

It has been reported that VTE is an important source of morbidity, death, and healthcare expenditures in individuals experiencing arthroplasty [9]. Also, although many strategies and guidelines have been used to reduce VTE, their efficacy still remains unclear [8]. Many authors advise pharmacological thromboprophylaxis, although, despite its use, 1.5% of patients experienced symptomatic VTE [10]. A prolonged aspirin protocol has been found to be at least as efficacious as rivaroxaban in averting VTE in individuals undergoing hip or knee arthroplasty. Therefore, aspirin can be deemed a safe and efficacious drug in the restraint of VTE following total-hip arthroplasty (THA) or total-knee arthroplasty (TKA) [11].

A level 3 of evidence study has shown that 1 in 71 individuals experiencing TKA and 1 in 167 individuals experiencing THA experienced VTE in the month following surgery. That study also showed that, over the past 8 years, the incidence percentages of VTE has decreased in TKAs, although the rates remained stable in THAs [9]. The effectiveness of enoxaparin in averting VTE in the TKAs cohort was substantially better than the efficacy of fondaparinux. In the THA cohort, the effectiveness of enoxaparin was substantially better than that of apixaban. The effectiveness of fondaparinux, dabigatran, rivaroxaban, apixaban, and aspirin were equivalent to that of enoxaparin in terms of decreasing mortality, serious bleeding, and complications associated with VTE. The study determined that all medications analyzed were non-inferior to enoxaparin [12].

### 2.4. TKA

In a study of individuals experiencing TKA, the percentage of VTE was significantly lower using DOACs than using enoxaparin. However, there were no substantial dissimilarities in terms of DVT, pulmonary embolism, and hemorrhagic complication rates [13]. Mian et al. investigated if tourniquet time, time to initiation of rivaroxaban (TTI_RIV_), or body mass index (BMI) were related to VTE after surgery. They observed substantial dissimilarities in tourniquet times between the VTE group and the non-VTE cohort, but not for TTI_RIV_ and BMI. Therefore, it was concluded that lengthy tourniquet utilization could be a possible risk factor for VTE after surgery [14]. In another study, no dissimilarity in clinically important major and non-major bleeding was observed between dabigatran and rivaroxaban during the 42 days after TKA [15].

### 2.5. THA 

In primary THA, aspirin and factor Xa inhibitors have yielded better DVT prophylaxis than enoxaparin and warfarin and with inferior percentages of anemia after surgery [16]. Gonzalez della Valle et al. have advocated multimodal prophylaxis [17] (Table 3). 

However, Gonzalez della Valle et al. stated that postoperative anticoagulation should be cautious, given that a small percentage of patients (2.5%) developed VTE or passed away from doubtful or established pulmonary embolism [17].

### 2.6. Orthopedic Surgery in Children

A thrombotic risk has been reported in children undergoing orthopedic surgery. However, it has also been stated that pharmacological thromboprophylaxis should not be employed, even in teenagers, and that adequate hydration and prompt mobilization (mechanical thromboprophylaxis) are usually sufficient to prevent VTE. In fact, the same authors mentioned that pharmacological prophylaxis should only be prescribed after analyzing the risk factors and the orthopedic background of each child, and that LMWHs are the most frequently employed thromboprophylactic drugs in children because of their excellent tolerance and effectiveness [18].

Table 4 summarizes all drugs that have been used for pharmacological thromboprophylaxis in the general population after orthopedic surgery [2,3,4,5,6,7,8,9,10,11,12,13,14,15,16,17,18].

## 3. Thromboprophylaxis in People with Hemophilia (PWH) Experiencing Orthopedic Surgery

A PubMed (MEDLINE) and Cochrane Library search of articles on thromboprophylaxis after orthopedic surgery in PWH was carried out. The keywords employed were “hemophilia thromboprophylaxis”. The chief inclusion criteria were articles focused on VTE after orthopedic surgery. Articles not focused on such a topic were dismissed. The requests were dated from the inception of the search engines until 31 July 2023. From the 94 articles (93 in PubMed, 1 in The Cochrane Library), those that were most directly related to the title of this paper were chosen (14 articles, 13 in PubMed, 1 in The Cochrane Library). This paper is not a systematic literature review, but a narrative review of the literature of the papers found in PubMed and The Cochrane Library that were deemed to be closely related to the title of this paper. Appendix A shows the flow chart used in this review regarding the role of pharmacological thromboprophylaxis after orthopedic surgery in PWH. Table 5 summarizes the information published to date on the role of pharmacological thromboprophylaxis in PWH undergoing orthopedic surgery [19,20,21,22,23,24,25,26,27,28,29,30,31].

## 4. Conclusions

This narrative review of the literature, consisting of papers found in PubMed and The Cochrane Library published up to 31 July 2023, analyzed those that were deemed to be closely related to the title of this paper. It was encountered that the peril of deep vein thrombosis (DVT) in people with hemophilia (PWH) after orthopedic surgery is very small, such that pharmacological thromboprophylaxis is not necessary in most cases.

However, based on the Caprini score, suggestions made in the literature for the general population are the following: patients must walk early and frequently after surgery; in individuals at very small risk of venous thromboembolism—VTE (Caprini score 0), no added prophylaxis is necessary; in individuals at small risk of VTE (Caprini score 1 to 2), mechanical [dynamic (intermittent plantar or pneumatic compression device), static (graduated compression stockings)], or pharmacological thromboprophylaxis [LMWH (low-molecular weight heparin) or fondaparinux subcutaneously, or oral dabigatran, rivaroxaban, or apixaban] is advised; in individuals at elevated risk of VTE (Caprini score 3 to 4) and in individuals at very elevated risk of VTE (Caprini score ≥ 5), pharmacological thromboprophylaxis alone or combined with mechanical prophylaxis should be used. However, in patients in whom the Caprini score has not yet been proven (orthopedic surgery), it is advisable to perform pharmacological thromboprophylaxis based on the risk factors of each individual and those specific to the surgical intervention. Generally, pharmacologic thromboprophylaxis should be prolonged until the patient is able to walk or is discharged from the hospital; however, certain orthopedic surgical procedures might require a longer duration of pharmacological thromboprophylaxis. 

The hemophilia literature states that the use of pharmacological thromboprophylaxis should only be carried out in PWH undergoing major orthopedic surgery (TKA, THA, ankle arthrodesis) who have added VTE risk factors such as old age, prior VTE, varicose veins, general anesthesia, cancer, factor V (Leiden) mutation, overweight, and treatment with the oral contraceptive pill (in women with von Willebrand’s illness). If we discover various risk factors for VTE in PWH who experience orthopedic surgery, we should conduct the same type of thromboprophylaxis that is advised for non-hemophilia individuals. However, the review of the literature on PWH has shown that only a few authors have used pharmacological prophylaxis with LMWH (subcutaneous enoxaparin) for a short period of time (10–14 days) in some patients who had risk factors for VTE. Only one group of authors used a small dose of DOAC in the dusk after surgery and the next day, specifically in individuals at elevated risk of VTE and elevated risk of bleeding after surgery.

In the field of thromboprophylaxis during orthopedic surgery in PWH, the problem of the lack of studies with a sufficient number of patients and adequate design to be able to make scientifically sound recommendations persists. This is due to the rarity of hemophilia and the decreasing need for orthopedic surgery due to primary hematological prophylaxis, which is practically eliminating cases of severe hemophilic arthropathy in developed countries such as Europe and USA. For future research, it would be necessary to carry out well-designed multicenter studies that could demonstrate, in a scientifically sound manner, whether pharmacological thromboprohylaxis is or is not necessary in hemophilic patients undergoing orthopedic surgery. Until this goal is achieved, it would be reasonable to use the same recommendations used in the general population, although most publications on hemophilia do not follow this line of reasoning. On the contrary, the publications to date on hemophilia simply advise early mobilization and mechanical methods as VTE prevention techniques. 

This article has two main limitations: the retrospective nature of the analysis and the variations in practices. It would be very recommendable to carry out a prospective study using the Caprini score-based algorithm. In addition, we cannot forget the potential underrepresentation of isolated instances of venous thrombosis in PWH experiencing arthroplasties, due to the retrospective nature of the study.

## Figures and Tables

**Table 1 jcm-12-05574-t001:** Caprini risk evaluation method for VTE in the general population.

1 Point for Each Risk Factor	2 Points for Each Risk Factor	3 Points for Each Risk Factor	5 Points for Each Risk Factor
Age from 41 to 60 years	Age from 61 to 74 years	Age over 75 years	Stroke (suffered less than one month ago)
Minor surgical intervention	Arthroscopy	History of VTE	Elective joint arthroplasty
Body mass index over 25 kg/m^2^	Major open surgery (of more than 45 min duration)	Family history of VTE	Fracture of the hip, pelvis, or leg
Swelling of the lower limbs	Laparoscopic surgery (of more than 45 min duration)	Factor V Leiden	Acute spinal cord injury (suffered less than one month ago)
Varicose veins	Cancer	Prothrombin 20210A	
Gestation or postpartum	Restricted to bed (for more than 72 h)	Lupus anticoagulant	
History of unexplained or repeating spontaneous abortion	Immobilization with a plaster cast	Anticardiolipin antibodies	
Treatment with oral contraceptives or hormone replacement	Use of a central venous access	Raised serum homocysteine	
Sepsis over the last month		Heparin-induced thrombocytopenia	
Severe lung illness, including pneumonia (over the last month)		Other congenital or acquired thrombophilia	
Abnormal pulmonary function			
Acute myocardial infarction			
Congestive heart failure (over the last month)			
History of inflammatory bowel illness			
Medical individual at bed rest			
Other risk factors			

**Table 2 jcm-12-05574-t002:** Recommendations of thromboprophylaxis in the general population (*).

Caprini score 0 (very small risk of VTE)	No additional prophylaxis is required.
Caprini 1 to 2 (small risk of VTE)	Mechanical prophylaxis [dynamic (intermittent plantar or pneumatic compression device), static (graduated compression stockings)], or both; or pharmacological prophylaxis (subcutaneous LMWH or fondaparinux, or oral dabigatran, rivaroxaban, or apixaban).
Caprini 3 to 4 (moderate risk of VTE)	Pharmacological thromboprophylaxis only or combined with mechanical prophylaxis.
Caprini ≥ 5 (elevated risk of VTE)	Pharmacological thromboprophylaxis only or combined with mechanical prophylaxis.

(*) Individuals should walk promptly and regularly after the surgical procedure. VTE, venous thromboembolism; LMWH, low-molecular weight heparin.

**Table 3 jcm-12-05574-t003:** Multimodal thromboprophylaxis for VTE in THA in the general population.

Discontinuation of procoagulant drugs
Grading of the risk of VTE
Regional anesthesia
Intravenous bolus of unfractionated heparin before femoral preparation
Quick mobilization
Chemoprophylaxis tailored to the patient’s risk of VTE

VTE, venous thromboembolism; THA, total-hip arthroplasty.

**Table 4 jcm-12-05574-t004:** Types of pharmacological thromboprophylaxis used in the literature for averting VTE after orthopedic surgery in the general population.

Aspirin (100 mg oral/24 h)
Vitamin K antagonists (oral Warfarin): begun the dusk after operation; the dose must be titrated to accomplish an international normalized ratio (INR) of 2–3
LMWHs, such as enoxaparin (40 mg subcutaneous/24 h)
DOACs, either thrombin inhibitors (dabigatran, 150 mg oral/12 h); or FXa inhibitors (rivaroxaban, 20 mg oral/24 h; and apixaban, 5 mg oral/24 h) or subcutaneous fondaparinux (2.5 mg/24 h subcutaneously)

VTE, venous thromboembolism; LMWH, low-molecular weight heparins; DOACS, direct oral anticoagulants; FXa, activated factor X.

**Table 5 jcm-12-05574-t005:** Data on pharmacological thromboprophylaxis of venous thromboembolism (VTE) in people with hemophilia (PWH).

Authors [Reference]	Year	Methods	Results	Conclusions
Pruthi et al. [19]	2000	Case report.	These authors reported a case of VTE in an individual with hemophilia B and factor V Arg506Gln (factor V Leiden).	In PWH with a family history of VTE, we must look for the presence of the factor V Arg506Gln (Leiden) mutation and other thrombophilias.
Ozelo [20]	2012	This review article stated that, for most PWH, the employment of graded compression stockings and prompt mobilization can be enough to avert VTE. The use of pharmacological thromboprophylaxis should be deemed just for PWH with important additional risk factors for VTE.	For women with von Willebrand illness treated with factor concentrates replacement therapy experiencing surgical procedures, factor VIII plasma levels should be checked and thromboprophylaxis should be deemed if any other thrombosis risk factor exists.	It is essential to determine risk evaluation instruments that can help to establish the most efficacious and safe practice to avert DVT in PWH and other bleeding diseases who experience surgical interventions.
Krekeler et al. [21]	2012	These authors analyzed the postoperative course in 85 PWH (mean age: 43 years) and 139 surgical procedures were studied. They chiefly consisted of major orthopedic surgery (58 TKAs, 15 THAs, 17 other major orthopedic surgery), but 15 minor orthopedic procedures were also studied. Other surgical procedures were abdominal-surgical (18), urological (8), neurosurgical (5).	None of the individuals suffered symptomatic DVT or pulmonary embolism.	There seemed to be a diminished risk of VTE in PWH after surgery
Rodríguez-Merchán [22]	2012	Review article.	Risk factors for VTE in PWH were orthopedic surgery, old age, prior VTE, varicose veins, general anesthesia, cancer, factor V (Leiden) mutation, overweight, and the use of oral contraceptive pills (in women with von Willebrand illness).	If we discover various risk factors for VTE in PWH who experience orthopedic surgery, we should perform the same type of pharmacological thromboprophylaxis advised for non-hemophilia individuals.
Siboni et al. [23]	2014	These authors analyzed 35 orthopedic interventions (6 minor and 29 major) performed in 22 PWH.	Pharmacological prophylaxis for VTE was conducted in only one case with no excessive hemorrhage.	Two individuals with hemophilia (2/22) not on pharmacological thromboprophylaxissuffered superficial thrombophlebitis.
Takedani et al. [24]	2015	These authors studied 38 TKA in 33 PWH with hemophilia utilizing US.	DVT was not detected.	The risk of DVT in PWH after TKA might be smaller than that in non-hemophilia individuals.
Hermans, et al. [25]	2016	These authors reported the outcomes of three independent prospective studies assessing, by systematic US-Doppler imaging, the prevalence of subclinical DVT in PWH sent for major orthopedic surgery. In total, 214 PWH experiencing 231 major orthopedic procedures of the lower limbs (136 TKAs, 49 ankle, and 34 hip surgeries) were analyzed.	Patients were not given pharmacological thromboprophylaxis. No case of clinical DVT or pulmonary embolism was found. Eleven cases of distal subclinical DVT involving one (5) or two (6) calf veins were encountered, of which six were managed with a small dose and a short period of LMWH. The prevalence of subclinical DVT was around 5%.	The risk of DVT after major orthopedic surgery was very small and pharmacological thromboprophylaxis was for most individuals not needed.
Buckner, et al. [26]	2016	These authors analyzed 46 PWH. Six participants (13%) were managed with bypassing agents during the perioperative period; the remaining forty participants were given factor VIII or IX replacement. Intermittent pneumatic compression devices were utilized after surgery in 23 participants (50%), and 4 participants (8.7%) were also given LMWH prophylaxis.	One participant (2.2%) suffered distal DVT six days after TKA. One participant (2.2%) suffered pulmonary embolism nine days after bilateral TKA. No participants presented asymptomatic DVT. Eighteen participants (39.1%) suffered major hemorrhage, and three participants (6.5%) suffered minor hemorrhage.	The detected incidence of US-detectable, asymptomatic DVT in PWH following TKA or THA in this study was small, but the prevalence of symptomatic VTE (4.3%) seemed similar to the calculated prevalence in the general population without pharmacological thromboprophylaxis.
Holderness, et al. [27]	2016	These authors analyzed 28 PWH who underwent 38 total joint arthroplasty procedures. LMWH was used in 29 procedures (76%) and was discontinued promptly in 3 interventions (2 individuals) due to non-joint hemorrhage.	Symptomatic VTE was not found.	No individual in this cohort suffered symptomatic VTE, whether or not pharmacological thromboprophylaxis was given.
Raza, et al. [28]	2016	These authors analyzed 23 PWH who experienced elevated-risk surgeries (THA and TKA).	VTE prophylaxis included compression devices in 52% of patients, and pharmacological thromboprophylaxis with enoxaparin in one patient (4%). Ten (43%) patients were not given VTE pharmacological thromboprophylaxis.	At 1-year follow-up, no evidence of clinical VTE was found in this series.
Ahmed, et al. [29]	2016	This review article recommended evaluation of individual risk for VTE, taking into account the type of the surgery, anesthesia, severity of bleeding disorder, age, BMI, history of DVT, and the presence of cancer and other elevated-risk illnesses. VTE risk should be balanced against the augmented risk of hemorrhage associated with the employment of anticoagulants in patients with known bleeding illnesses.	In PWH undergoing major surgery, these authors advised against regular postoperative employment of pharmacological thromboprophylaxis, especially for individuals with hemophilia A and B.	For PWH at an elevated risk of VTE and with an elevated bleeding risk after surgery, these authors deemed that giving a small dose of direct oral anticoagulant on the dusk after surgery and on the next day after surgery was appropriate.
Verstraete, et al. [30]	2020	These authors analyzed 46 PWH that experienced 67 orthopedic procedures. The majority (89.5%) were carried out with continuous infusion of clotting factor concentrates. Rehabilitation was begun on the first day after surgery.	We found 5 cases of subclinical DVT, all of them distal. Two individuals were given a short period (10–14 days) of LMWH. The prevalence of DVT was 7.5%.	Pharmacological thromboprophylaxis in PWH was probably not needed for the majority of individuals.
Ono and Takedani [31]	2020	These authors analyzed 11 TKAs in 11 PWH without a history of inhibitor. A pneumatic compression device was employed from the start of the surgical procedure until the individual could conduct standing exercises. US and contrast-enhanced CT of the lower extremities were performed. D-dimer was also measured.	DVT was not encountered by US, but contrast-enhanced CT discovered DVT in two individuals.	Contrast-enhanced CT discovered DVT in 18% of PWH who experienced TKA.

DVT, deep vein thrombosis; TKA, total-knee arthroplasty; THA, total-hip arthroplasty; US, ultrasound; LMWH, low-molecular weight heparin; THA, total-hip arthroplasty; BMI, body mass index; CT, computed tomography.

## Data Availability

Not applicable.

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
