# Peer review of "Pharmacological Thromboprophylaxis in People with Hemophilia Experiencing Orthopedic Surgery: What Does the Literature Say in 2023?"

_jcm, 2023, doi:10.3390/jcm12175574_

Round 1
Reviewer 1 Report
-
1)It's important to acknowledge the limitations-the retrospective nature of the analysis and the variations in practices and adjust the conclusions .
-
2)A prospective study using the Caprini score-based algorithm could be a valuable next step.
-
3)The potential underrepresentation of isolated instances of venous thrombosis in Hemophiliacs undergoing arthroplasties due to the retrospective nature of the study can be discussed .
-
4) Parts of conclusions can be toned down perhaps due to the lack of robust evidence from the review
-
-
Minor language editing advised .
Author Response
Pharmacological thromboprophylaxis in people with hemophilia experiencing orthopedic surgery: what does the literature say in 2023?
Dear Editor-in-Chief,
Below and in the revised manuscript (REVISION-1) you can see IN RED the changes related to Reviewer-1.
REVIEWER-1
Comments and suggestions to authors
1) It's important to acknowledge the limitations-the retrospective nature of the analysis and the variations in practices and adjust the conclusions.
AUTHOR: I have added the following sentences at the end of the “Conclusions”:
This article has two main limitations: the retrospective nature of the analysis and the variations in practices.
2) A prospective study using the Caprini score-based algorithm could be a valuable next step.
AUTHOR: I have added the following sentences at the end of the “Conclusions”:
It would be very recommendable to carry out a prospective study using the Caprini score-based algorithm.
3) The potential underrepresentation of isolated instances of venous thrombosis in Hemophiliacs undergoing arthroplasties due to the retrospective nature of the study can be discussed.
AUTHOR: I have added the following sentences at the end of the “Conclusions”:
Besides, it cannot be forgotten the potential underrepresentation of isolated instances of venous thrombosis in PWH experiencing arthroplasties due to the retrospective nature of the study.
4) Parts of conclusions can be toned down perhaps due to the lack of robust evidence from the review
AUTHOR: I have tried to tone down parts of the “Conclusions” the best I could.
Reviewer 2 Report
Thank you for your work!
The author draws attention to a little-studied problem (pharmacological thromboprophylaxis in people with hemophilia), collected in a review known data on this, which deserves respect. However, the review does not provide competent conclusions regarding the data being reviewed. Please transform the review with your findings, it will be greatly appreciated.
The comments include requirements and recommendations.
1) Minor mistakes:
‘HOspital-Idipaz’ (line 6) – erroneous upper case;
‘…subcutaneously] are…’ (line 42) – no opening square bracket;
‘…according to my criteria…’ (lines 148-149), ‘Figure 1 shows the flow chart of my quest approach…’ (lines 149-150) – it is better not to mention yourself in the text, instead you can use "used in this review".
2) Table 1 (line 66):
Place the table immediately after it is mentioned in the text (line 57);
‘…venous thromboembolism (VTE)…’ (line 66) – you have already deciphered this abbreviation, choose to use either the abbreviation or its decoding;
‘If yes: Type’ – what does it mean? Remove it or explain what it means;
‘SUBTOTAL’ and ‘TOTAL RISK FACTOR SCORE’ – this looks redundant, delete it or explain what it means.
2) Table 2 (line 68):
Place the table immediately after it is mentioned in the text (line 59);
‘Table 2. Recommendations of thromboprophylaxis in the general population: individuals should walk promptly and regularly after the surgical procedure’ (lines 68-69) – it's more like an output than a table name. Such statements can be placed in the text. Correct the table name.
3) ‘Total knee arthroplasty (TKA)’ (line 103), ‘…direct oral anticoagulants (DOACs)…’ (line 105), ‘Total hip arthroplasty (THA)’ (line 114), ‘deep vein thrombosis (DVT)’ (lines 280-281), ‘people with hemophilia (PWH)’ (line 281): You have already deciphered these abbreviations, choose to use either the abbreviation or its decoding.
4) ‘Some authors have advocated multimodal prophylaxis….’ (line 117): You didn't list the authors, and the table you link to doesn't have links to articles either. For the validity of the statement, add a link to the articles.
5) Table 3 (line 121): Place the table immediately after it is mentioned in the text (line 117).
6) Table 4 (line 134):
‘…pharmacological thromboprophylaxis used in the literature…’ – there are no references to studies, they should be added to the table.
7) Figure 1 (line 152):
The drawing is uninformative and occupies a large space, drawing attention to itself. Place this drawing in the Supplementary.
Even within 2023 (mentioned in the title of the review), the data on the number of articles may change. It is advisable to indicate the date of analysis of publications for which your data is true.
8) ‘hemophilia A and B’ (line 176): It would be good to discuss these two states in Introduction. You mention hemophilia in key words but don't discuss it at all in Introduction.
9) Table 5 (line 274):
The text (lines 155-271) before Table 5 is meaningless. The table looks much more informative and meaningful. Convert or remove the text (lines 155-271) before the table.
10) ‘Reviews offer a comprehensive analysis of the existing literature within a field of study, identifying current gaps or problems. They should be critical and constructive and provide recommendations for future research’ [https://www.mdpi.com/journal/jcm/instructions]:
The review does not have one of the most important aspects – the processing of the reviewed literature. This is missing neither in Abstract (line 8) nor in Conclusions (line 279). Please complete the review with competent conclusions.
Author Response
Pharmacological thromboprophylaxis in people with hemophilia experiencing orthopedic surgery: what does the literature say in 2023?
Dear Editor-in-Chief,
Below and in the revised manuscript (REVISION-1) you can see IN BLUE the changes related to Reviewer-2.
REVIEWER-2Principio del formulario
Comments and Suggestions for Authors
The author draws attention to a little-studied problem (pharmacological thromboprophylaxis in people with hemophilia), collected in a review known data on this, which deserves respect. However, the review does not provide competent conclusions regarding the data being reviewed. Please transform the review with your findings, it will be greatly appreciated.
The comments include requirements and recommendations.
1) Minor mistakes:
‘HOspital-Idipaz’ (line 6) – erroneous upper case;
AUTHOR: The mistake has been amended.
‘…subcutaneously] are…’ (line 42) – no opening square bracket;
AUTHOR: The mistake has been amended.
‘…according to my criteria…’ (lines 148-149), ‘Figure 1 shows the flow chart of my quest approach…’ (lines 149-150) – it is better not to mention yourself in the text, instead you can use "used in this review".
AUTHOR: It has been changed.
2) Table 1 (line 66):
Place the table immediately after it is mentioned in the text (line 57);
AUTHOR: Table 1 has been placed after it is mentioned in the text.
‘…venous thromboembolism (VTE)…’ (line 66) – you have already deciphered this abbreviation, choose to use either the abbreviation or its decoding;
AUTHOR: We have changed to VTE.
‘If yes: Type’ – what does it mean? Remove it or explain what it means;
‘SUBTOTAL’ and ‘TOTAL RISK FACTOR SCORE’ – this looks redundant, delete it or explain what it means.
AUTHOR: It has been deleted.
2) Table 2 (line 68):
Place the table immediately after it is mentioned in the text (line 59);
AUTHOR: Table 2 has been placed immediately after it is mentioned in the text.
Table 2. Recommendations of thromboprophylaxis in the general population: individuals should walk promptly and regularly after the surgical procedure’ (lines 68-69) – it's more like an output than a table name. Such statements can be placed in the text. Correct the table name.
AUTHOR: Statements have been placed at the bottom of the table. The table name has been corrected.
3) ‘Total knee arthroplasty (TKA)’ (line 103), ‘…direct oral anticoagulants (DOACs)…’ (line 105), ‘Total hip arthroplasty (THA)’ (line 114), ‘deep vein thrombosis (DVT)’ (lines 280-281), ‘people with hemophilia (PWH)’ (line 281): You have already deciphered these abbreviations, choose to use either the abbreviation or its decoding.
AUTHOR: Abbreviation has been used.
4) ‘Some authors have advocated multimodal prophylaxis….’ (line 117): You didn't list the authors, and the table you link to doesn't have links to articles either. For the validity of the statement, add a link to the articles.
AUTHOR: I meant Gonzalez della Valle et al [17]. It has been changed.
5) Table 3 (line 121): Place the table immediately after it is mentioned in the text (line 117).
AUTHOR: Table 3 has been placed immediately after it is mentioned in the text.
6) Table 4 (line 134):
‘…pharmacological thromboprophylaxis used in the literature…’ – there are no references to studies, they should be added to the table.
AUTHOR: At the end of the sentence I have added the references as follows:
Table 4 summarizes all drugs that have been used for pharmacological thromboprophylaxis in the general population after orthopedic surgery [2-18].
7) Figure 1 (line 152):
The drawing is uninformative and occupies a large space, drawing attention to itself. Place this drawing in the Supplementary.
Even within 2023 (mentioned in the title of the review), the data on the number of articles may change. It is advisable to indicate the date of analysis of publications for which your data is true.
AUTHOR: I have placed the drawing (Figure 1) in the Supplementary. The date of analysis of publications was 31 July 2023.
8) ‘hemophilia A and B’ (line 176): It would be good to discuss these two states in Introduction. You mention hemophilia in key words but don't discuss it at all in Introduction.
AUTHOR: I have included the following paragraph at the beginning of the “Introduction”
In 2023 Morfini et al stated that according to the Italian registry of surgery, hemophilia B individuals experience total joint arthroplasty less frequently than hemophilia A individuals. Besides, they updated the hematological replacement therapy required for surgery utilizing continuous infusion (CI) versus bolus injection (BI) of standard and extended half-life Factor VIII (FVIII) concentrates in individuals with severe hemophilia A. They mentioned two important complications of surgery: venous thromboembolism (VTE) and the risk of development of inhibitors and. Regarding primary hematological prophylaxis, for young hemophilia A patients they recommended to use it three times or twice weekly, even with standard half-life (SHL) recombinant FVIII (rFVIII) concentrates. For patients with severe or moderate hemophilia B they advised once weekly [1. MORFINI et al, 2023 – NEW REFERENCE].
9) Table 5 (line 274):
The text (lines 155-271) before Table 5 is meaningless. The table looks much more informative and meaningful. Convert or remove the text (lines 155-271) before the table.
AUTHOR: I have removed the text (lines 155-271).
10) ‘Reviews offer a comprehensive analysis of the existing literature within a field of study, identifying current gaps or problems. They should be critical and constructive and provide recommendations for future research’ [https://www.mdpi.com/journal/jcm/instructions]:
The review does not have one of the most important aspects – the processing of the reviewed literature. This is missing neither in Abstract (line 8) nor in Conclusions (line 279). Please complete the review with competent conclusions.
AUTHOR: I have added the following paragraph at the end of the “Conclusions”
In the field of thromboprophylaxis during orthopedic surgery in PWH, the problem of the lack of studies with a sufficient number of patients and adequate design to be able to make scientifically sound recommendations persists. This is due to the rarity of hemophilia and the decreasing need for orthopedic surgery due to primary hematological prophylaxis, which is practically eliminating cases of severe hemophilic arthropathy in developed countries such as Europe and USA. For future research it would be necessary to carry out well-designed multicenter studies that could demonstrate in a scientifically sound manner whether pharmacological thromboprohylaxis is or is not necessary in hemophilic patients undergoing orthopedic surgery. Until this goal is achieved, it would be reasonable to use the same recommendations used in the general population, although most publications on hemophilia do not follow this line of reasoning. On the contrary, the publications to date on hemophilia simply advise early mobilization and mechanical methods as VTE prevention techniques.
I have added the following paragraph at the beginning of the “Abstract” and at the beginning of the “Conclusions”
This narrative review of the literature of the papers found in PubMed and The Cochrane Library using the search engines until 31 July 2023 analyzed those that were deemed to be closely related to the title of this paper. It was encountered that the peril of deep vein thrombosis (DVT) in people with hemophilia (PWH) after orthopedic surgery is very small, such that pharmacological thromboprophylaxis is not necessary in most cases.
Round 2
Reviewer 2 Report
Dear author, I agree with the way you corrected the manuscript according to the comments. There are just minor comments I should address before the article is accepted.
1) At the beginning of the "Introduction", in the first paragraph you've included, there must be a missing part: the end of the sentence is absent (line 39) - something should be placed after "the risk of development of inhibitors and.". Additionally, the phrase "development of inhibitors" in this sentence looks awkuard. Perhaps, you meant something like "influence/impact/effect of inhibitors"?
2) Nothing is written for the following:
Funding: (line 229)
Data Availability Statement: (line 230)
Conflicts of Interest: (line 231).
If the first two statements are irrelevant - you could dismiss them. But for the "Conflicts of Interest" you have to mention that there are none of them.
Author Response
23 August 2023
Dear Reviewer,
Below you can see our responses to your comments and the changes made in the manuscript (REVISION-2) IN RED color.
REVIEWER-2 (SECOND ROUND)
Dear author, I agree with the way you corrected the manuscript according to the comments. There are just minor comments I should address before the article is accepted.
1) At the beginning of the "Introduction", in the first paragraph you've included, there must be a missing part: the end of the sentence is absent (line 39) - something should be placed after "the risk of development of inhibitors and.". Additionally, the phrase "development of inhibitors" in this sentence looks awkuard. Perhaps, you meant something like "influence/impact/effect of inhibitors"?
AUTHOR: I have changed the sentence as follows (IN RED)
They mentioned two important complications of surgery: venous thromboembolism (VTE) and the impact of inhibitors.
2) Nothing is written for the following:
Funding: (line 229)
Data Availability Statement: (line 230)
Conflicts of Interest: (line 231).
If the first two statements are irrelevant - you could dismiss them. But for the "Conflicts of Interest" you have to mention that there are none of them.
AUTHOR: I have included the following regarding “Funding” and “Conflicts of Interest” IN RED). The other statement “Data Availablity Statement” is irrelevant (it has been dismissed).
Funding: This article has not been funded.
Conflicts of Interest: The author declares that he has no conflicts of interest.